# Molecular basis for azetidine-2-carboxylic acid biosynthesis

Tim J. Klaubert [1], Jonas Gellner[1,2,11], Charles Bernard [3,10,11], Juliana Effert [4,11], Carine Lombard[3], Ville R. I. Kaila [2], Helge B. Bode[4,5,6,7,8,9], Yanyan Li [3] ✉ & Michael Groll [1] ✉

Azetidine-2-carboxylic acid (AZE) is a long-known plant metabolite. Recently, AZE synthases have been identified in bacterial natural product pathways involving non-ribosomal peptide synthetases. AZE synthases catalyse the intramolecular 4-exo-tet cyclisation of *S*-adenosylmethionine (SAM), yielding a highly strained heterocycle. Here, we combine structural and biochemical analyses with quantum mechanical calculations and mutagenesis studies to reveal catalytic insights into AZE synthases. The cyclisation of SAM is facilitated by an exceptional substrate conformation and supported by desolvation effects as well as cation-π interactions. In addition, we uncover related SAM lyases in diverse bacterial phyla, suggesting a wider prevalence of AZE-containing metabolites than previously expected. To explore the potential of AZE as a proline mimic in combinatorial biosynthesis, we introduce an AZE synthase into the pyrrolizixenamide pathway and thereby engineer analogues of azabicyclenes. Taken together, our findings provide a molecular framework to understand and exploit SAM-dependent cyclisation reactions.

*S*-adenosylmethionine (SAM)-dependent enzymes have a remarkable versatility in catalysing chemical transformations due to the intrinsic electrophilicity of the sulfonium group[1–4]. One plausible reaction is cyclisation of the aminocarboxypropyl moiety by an intramolecular nucleophilic attack of the Cγ-carbon, yielding homoserine-γ-lactone (HSL), 1-aminocyclopropane-1-carboxylic acid (ACC) or azetidine-2-carboxylic acid (AZE), with the concomitant release of methylthioadenosine (MTA, Fig. 1a). While the corresponding HSL and ACC synthases have been extensively studied[5–7], the enzymes responsible for AZE production from SAM remained unidentified until recently[8,9]. In this regard, AZE has long been recognised as a plant metabolite playing a crucial role

in the defence mechanisms to protect against herbivores, competitors and phytopathogens[10,11]. As an analogue of proline, AZE can be incorporated into proteins, which results in altered stability and conformation leading to the disruption of protein function[12–14].

AZE and its derivatives are rare components of plant and microbial natural products[15]. These include plant phytosiderophores such as nicotianamine[16], azetidomonamides in *Pseudomonas aeruginosa*[8,17], vioprolides in *Cystobacter violaceus*[9], clipibicylene and azabicyclenes in *Streptomyces cattlya*[18] as well as bonnevillamides in *Streptomyces* sp. GSL-6B and UTZ13[19,20] (Fig. S1). Plant nicotianamine synthases catalyse the assembly of three aminocarboxypropyl moieties from SAM

[1]Center for Protein Assemblies, Department Bioscience, School of Natural Sciences, Technical University Munich, Garching, Germany. [2]Department of Biochemistry and Biophysics, Stockholm University, Stockholm, Sweden. [3]Laboratory Molecules of Communication and Adaptation of Microorganisms (MCAM), UMR 7245 CNRS-MNHN (Muséum National d'Histoire Naturelle), Paris, France. [4]Max Planck Institute for Terrestrial Microbiology, Department of Natural Products in Organismic Interactions, Marburg, Germany. [5]Institute of Molecular Biosciences, Goethe University Frankfurt, Frankfurt am Main, Germany. [6]Myria Biosciences AG, Basel, Switzerland. [7]Department of Chemistry, Philipps-University Marburg, Marburg, Germany. [8]LOEWE Centre for Translational Biodiversity Genomics (LOEWE TBG) & Senckenberg Gesellschaft für Naturforschung, Frankfurt am Main, Germany. [9]Center for Synthetic Microbiology (SYNMIKRO), Phillips University Marburg, Marburg, Germany. [10]Present address: Institut Pasteur, CNRS UMR3525, Microbial Evolutionary Genomics, Paris, France. [11]These authors contributed equally: Jonas Gellner, Charles Bernard, Juliana Effert. ✉e-mail: yanyan.li@mnhn.fr; michael.groll@tum.de

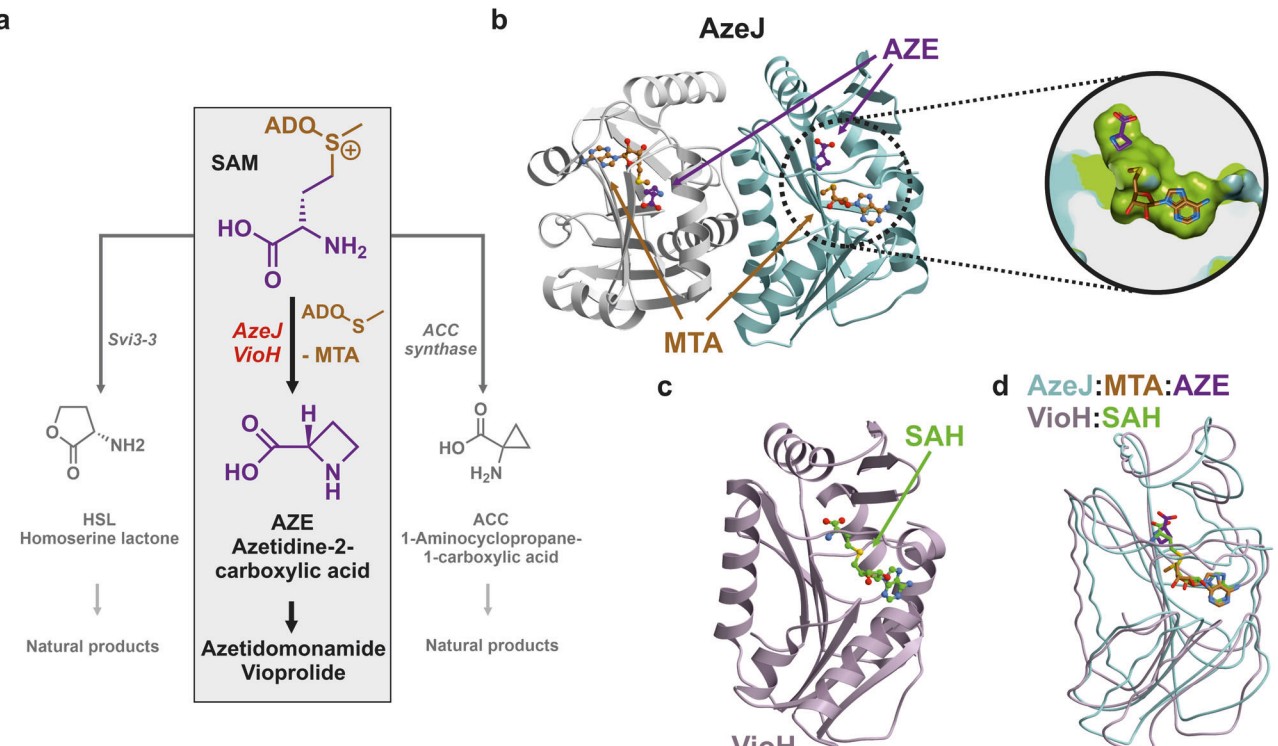

**Fig. 1 | X-ray structures of AzeJ and VioH. a** Intramolecular cyclisation of *S*-adenosylmethionine (SAM) can result in three-, four-, or five-membered ring products[1]. Enzymes catalysing 1-aminocyclopropane-1-carboxylic acid (ACC), azetidine-2-carboxylic acid (AZE), and homoserine lactone (HSL) include aminocyclopropane-1-carboxylic acid (ACC) synthase, AzeJ/VioH and Svi3-3, respectively. **b** Ribbon drawing of the AzeJ homodimer (PDB ID: 8RYE) bound to 5′-methylthioadenosine (MTA, brown) and AZE (purple). The close-up provides a cut open view of the active site surface (carbons in green) in complex with MTA and AZE (sticks). **c** Ribbon drawing of monomeric VioH (PDB ID: 8RYG) bound to *S*-adenosyl-*L*-homocysteine (SAH, green). **d** The overlay of AzeJ:MTA:AZE (turquoise) and VioH:SAH complex structures (light purple) indicates their structural similarity.

and the cyclisation of the terminal one to AZE[21]. However, the molecular details of cyclisation remain unknown. More recently, AZE synthases have been discovered in bacteria, namely AzeJ and VioH, which are involved in the biosynthesis of azetidomonamides and vioprolides, respectively[8,22]. Interestingly, they show sequence similarity to the class I methyltransferases (MT1). The molecular basis that causes the shift in reaction trajectory is hence intriguing. Similar proteins have also been found to be encoded by the biosynthetic gene clusters (BGCs) of clipibyclenes[18] and bonnevillamides[20]. In these pathways, the resulting AZE or methyl-AZE, as in the case of vioprolides and bonnevillamides, serves as a building block and is incorporated into the peptide backbones by specialised non-ribosomal peptide synthetases (NRPSs).

Taken together, the discovery of AZE synthases opens up avenues for investigating the molecular principles underlying SAM-mediated intramolecular cyclisation. Our present study raises fundamental questions regarding the divergence of catalytic functions within the MT1 family, as well as the natural occurrence and role of AZE. Considering the potential of the strained azetidine motif as a target for drug development[11], the combination of AZE synthases with proline-incorporating NRPS pathways holds promise for the creation of innovative molecules. In this work we provide a detailed analysis of the structural, biochemical and phylogenetic aspects of AzeJ and VioH. Additionally, we showcase the biotechnological value of AzeJ in combinatorial biosynthesis.

## Results
### Overall structures of AzeJ and VioH
To gain a better understanding of the intramolecular cyclisation reaction of SAM to MTA and AZE, we determined the high-resolution

structures of AzeJ and VioH. To this end, recombinant proteins were heterologously produced in *Escherichia coli*, purified by Ni²⁺ affinity and size-exclusion chromatography and crystallised (see Supplementary Information, Tables S1–S3 and Fig. S2). Proteins without ligand were recalcitrant to crystallisation. In contrast, we were able to obtain crystals in the presence of SAM or *S*-adenosyl-*L*-homocysteine (SAH), which diffracted up to 1.95 Å. Our analyses revealed that AzeJ forms a homodimer, whilst VioH is monomeric (Fig. 1 and Fig. S3). Although the sequence identity between AzeJ and VioH is only 28% (Fig. S4), their monomers share a remarkable structural similarity (backbone root mean square deviation (rmsd) = 1.8 Å, 72% Cᵅ-atoms). Each subunit adopts a classic Rossmann fold, consisting of a 7-stranded β-sheet (β3↑, β2↑, β1↑, β4↑, β5↑, β10↓, β9↑) surrounded by helices α4-6, α8-9 and α11 (Figs. S5, S6), a topology that is typical for MT1 enzymes[2]. A structure homology search of AzeJ using the DALI server[23] indicates a strong relationship with other SAM-dependent MT1s, with a close structural similarity to 1,6-didemethyltoxoflavin (DDMT)-N1-MT from *Burkholderia thailandensis* (PDB ID: 5UFM, Z-score = 19.2; sequence identity 19%, Fig. S7a)[24]. However, a significant difference is observed in the arrangement of the N-termini between AzeJ and DDMT-N1-MT, whereas the respective compact α/β sandwich domains match perfectly.

Previous reports have shown that AzeJ and VioH differ in the synthesis of AZE[8,22], with VioH having only a low activity (Fig. S8). It was further speculated that VioH may require a protein partner, *i.e.*, a radical SAM (rSAM) methyltransferase (VioG), to be fully functional, since this pair of enzymes produces methyl-AZE in vivo[22]. Nevertheless, the catalysis in AzeJ and VioH would follow the same mechanistic principles, given their structural similarity. We have therefore focused our current study on AzeJ.

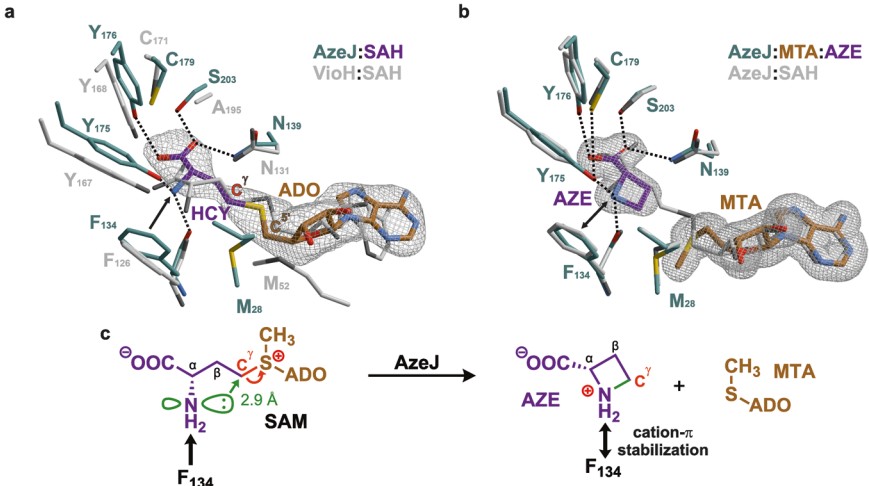

**Fig. 2 | Molecular insights into the active sites of AzeJ and VioH. a** Close-up of the catalytic centre of AzeJ:SAH (PDB ID: 8RYD), highlighting the specificity pocket (residues in cyan, adenosyl residue (ADO) in brown, homocysteine (HCY) in purple). The VioH:SAH complex structure is superimposed (PDB ID: 8RYG, C-atoms in grey). Residues are numbered according to the primary sequence of the respective enzyme. In AzeJ, the activation of the NH₂ group of HCY via F134 π-interactions is shown by a black arrow. **b** Crystallisation of AzeJ in the presence of SAM resulted in the AzeJ:MTA:AZE complex (PDB ID: 8RYE, MTA in brown, AZE in purple). The amine group of AZE forms hydrogen bonds with the hydroxyl group of Tyr175 and

the carbonyl oxygen of Phe134. Strong cation-π interactions with Phe134 (black double arrow) indicate a protonated state of the nitrogen atom. The overlay with AzeJ:SAH (grey) illustrates catalysis without structural rearrangements. The $2F_o$-$F_c$ electron density maps (1σ, grey mesh) are displayed for ligands that were omitted for phasing. **c** Proposed catalytic mechanism of AzeJ. *Left panel:* The electrophilic $C^\gamma$-carbon (red) is attacked by the amine (distance 2.9 Å). *Right panel:* The positive charge is transferred from SAM to the AZE nitrogen atom and stabilised by cation-π interactions with Phe134 (black double arrow).

## Substrate/product binding

The complex structure of AzeJ (PDB ID: 8RYD, 3.0 Å, Table S4) with SAH depicts the surrogate well defined in the $F_o$-$F_c$ electron density map. Interestingly, the ligand binding is accompanied by structural rearrangements of the N-terminus (see below), resulting in a closed state that shields SAH from the bulk solvent. The surrogate is located at the C-terminal ends of β1 and β2, with the binding pocket enclosed by three short helices α1-3, three β-sheets β6-8 and helix α10 (Fig. S6). The SAH-adenine moiety is coordinated by a GxxxG motif at β1, while MT1s usually use a conserved GxGxG motif for nucleobase binding[2]. The SAH ribose is hydrogen-bonded to Asp85 at the end of β2, and the homocysteine (HCY) moiety of SAH adopts a kinked conformation. Moreover, the X-ray structure illustrates that HCY α-amine is bound to AzeJ in its neutral form. The comparison of the active sites between AzeJ and DDMT-N1-MT (Fig. S7b) reveals key insights into substrate orientation and binding specificity. In AzeJ, HCY adopts a similar orientation to the substrate DDMT during methylation in DDMT-N1-MT, suggesting a conserved binding mode between these two enzymes. However, while DDMT-N1-MT accommodates a linear conformation of SAH, this arrangement is blocked in AzeJ by specific protein interactions involving the $C^\gamma$-carbon of HCY and the $C^{5'}$-methylene group of the ribose moiety. Furthermore, the coordination of a linear SAH in AzeJ is compromised by the absence of a counterion to stabilise the carboxylate of HCY (Fig. S7b). HCY-COO⁻ is hydrogen-bonded to Tyr176, Asn139 and Ser203, while HCY-NH₂ coordinates with the carbonyl oxygen of Phe134 and the hydroxyl group of Tyr175. Moreover, Phe134 engages a π interaction with HCY-NH₂[25–27], which would increase the nucleophilicity of the latter[28]. The kinked conformation directs the nucleophilic nitrogen towards the $C^\gamma$-carbon of HCY (2.9 Å, Fig. 2a) and induces the cyclisation reaction. In addition, we have also obtained a VioH structure in complex with SAH at 2.0 Å resolution (PDB ID: 8RYG, Table S4), whose active site is similar to that of AzeJ (Fig. 2a).

To obtain further insight into the substrate recognition process, we determined two additional, high-resolution AzeJ structures co-crystallised with SAM (PDB ID: 8RYE, at 1.95 Å ($P2_12_12$); PDB ID: 8RYF, at 1.95 Å ($P4_222$), Table S4). Interestingly, the $F_o$-$F_c$ electron density maps

reveal that MTA and AZE are bound to the active site, suggesting that the substrate has been converted via an intramolecular nucleophilic nitrogen attack on the $C^\gamma$-carbon. Intriguingly, the structural superposition of AzeJ:SAH with AzeJ:MTA:AZE only shows marginal changes of the four-membered AZE-ring in the active site (rmsd = 0.3 Å, 221 $C^\alpha$-atoms), except that the nitrogen and $C^\gamma$-atoms are moved by 1.5 Å towards each other (Fig. 2b). The amine group of AZE forms hydrogen bonds with the hydroxyl group of Tyr175 and the carbonyl oxygen of Phe134. Strong cation-π interactions with Phe134 indicate a protonated state of the nitrogen atom in the AZE product. Based on the atomic insights, we propose that during cyclisation the positive charge of the sulfonium group migrates to the C-N bond and accumulates on the AZE amine. This charge distribution is stabilised by cation-π interactions with Phe134 (Fig. 2b, c), thereby facilitating the subsequent cleavage of the $C^\gamma$-S bond, which ultimately forms the neutral molecule MTA. Moreover, the AzeJ:MTA:AZE structure provides insights into how the α-amine of SAM, protonated under physiological conditions (SAM-NH₃⁺), can be deprotonated. Similarly as seen in AzeJ:SAH, the carboxylate of AZE forms three hydrogen bonds with Asn139, Y176 and S203 (Figs. S9 and S10a). The latter two residues further coordinate with Arg172, which is additionally H-bonded to Thr193 and a water molecule (Fig. S9). This intricate hydrogen bonding network could potentially facilitate the transfer of the proton of SAM-NH₃⁺ to the bulk solvent.

Next, we modelled the AzeJ:SAM complex by adding an *S*-configured methyl side chain to the sulphur of SAH in the AzeJ:SAH surrogate structure (Fig. S10b). The model suggests that the methyl group is stabilised by van der Waals interactions with Leu136. On the other hand, the positive charge of the sulfonium group in SAM could be polarised by cation-π interactions with the aromatic side chain of Tyr18 (Fig. S10b). This coordination could enhance the interplay between Tyr18OH and Asn139NH₂, thereby increasing substrate affinity by coordinating the carboxylate of SAM. We thus conclude, that this interaction may involve a structural rearrangement of the N-terminus, changing AzeJ from an open to a closed, solvent-shielded state to prevent side reactions with water molecules during catalysis. Once MTA and AZE are formed, the positive charge of the sulfonium group is

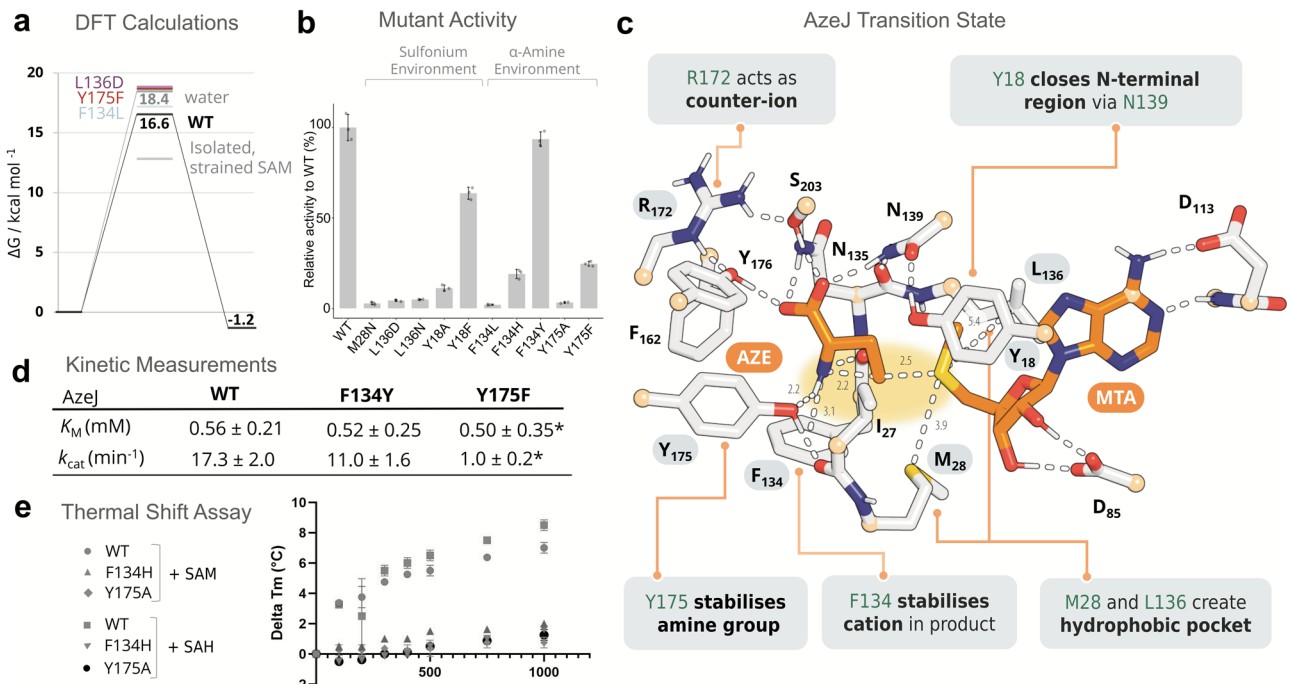

**Fig. 3 | Computational and biochemical characterisation support the proposed mechanism of AzeJ catalysis. a** Electronic energies of reactant (SAM), transition and product states (AZE + MTA) for AzeJ wild-type (WT) and mutants based on DFT cluster models of the active site (Fig. S12), calculated at the B3LYP-D3/def2-TZVP/ $\varepsilon = 4$ level. Comparative reactions in solvent were modelled using implicit solvation (in water, $\varepsilon = 80$; hydrophobic, $\varepsilon = 4$). More details are provided in the Methods section and Table S5. **b** Relative activity of AzeJ variants compared to WT. Data are presented as mean values ± standard deviation (SD) with three replicates. Experiments are repeated twice independently. **c** DFT model of the AzeJ WT transition state (alternative view in Fig. S16). Highlighted residues were analysed by mutagenesis; atoms shown in light orange were fixed during geometry and reaction pathway optimisations. For clarity, only polar hydrogens are shown (distances in Å). **d** Determination of kinetic parameters ($K_M$, $k_{cat}$, *denotes apparent parameters without consideration of substrate inhibition) (graphs see Fig. S19). **e** Evaluation of ligand binding by thermal shift assays for AzeJ WT, F134Y and Y175F variants. Data are presented as mean values ± SD with two biological replicates. Source data are provided as a Source Data file for (**b**, **d** and **e**).

transferred to the amine group of AZE and the sulphur atom of MTA no longer coordinates with Tyr18 (Fig. S10a). Consequently, the interaction between Tyr18OH, Asn139NH$_2$ and the AZE-carboxylate is weakened, which could be associated with the transition of AzeJ to an open state. This model would imply a function of Tyr18 as a gating residue. The conformational dynamics of AzeJ between open and closed states during catalysis were supported by structural comparison between AzeJ:MTA and AzeJ in complex with both MTA and AZE (Fig. S11).

**Exploring AzeJ catalysis by quantum mechanical calculations**

To gain insight into the mechanistic principles of AzeJ catalysis and to estimate the reaction energies for the conversion of SAM to MTA and AZE, we applied density functional theory (DFT) calculations. DFT cluster models of the active site were constructed based on the AzeJ:SAH crystal structure (see Methods and Fig. S12). The product state was modelled by adjusting the positions of AZE atoms in accordance to the AzeJ:MTA:AZE complex. While the α-amine of SAM is generally protonated under physiological conditions, a neutral amine state is necessary for bonding to the C$^\gamma$-carbon. To correctly capture the protonation states and gain insight into the deprotonation process, we tested if the proton could be transferred to SAM carboxylate. Although this transfer is energetically feasible (uphill by ca. 9 kcal mol$^{-1}$), the subsequent formation of AZE-COOH is precluded by the geometry and high energy of the product (Fig. S13 and Table S5). We therefore hypothesise that upon binding to AzeJ, the proton is transferred from SAM-NH$_3^+$ to the bulk solvent—either directly in the open state or via Tyr175 or Tyr176—and modelled SAM with a neutral α-amine accordingly (Fig. S12 and Table S5).

Next, the conversion to AZE was modelled as a single transition state (TS) S$_N$2 reaction, as outlined in the Methods section. We

obtained a kinetically feasible barrier of 16.6 kcal mol$^{-1}$, corresponding to a rate of ~13 s$^{-1}$ according to transition state theory (Fig. 3a)[29]. Notably, the barrier is lowered compared to deprotonated SAM in water (Fig. 3a), demonstrating that AzeJ catalysis goes beyond simple deprotonation. Remarkably, AzeJ also balances the energies of the reactant SAM (sulfonium group) and the product AZE (strained azetidine ring), with a marginal difference of about 1 kcal mol$^{-1}$ (Fig. 3a). Further analysis revealed that AzeJ stabilises the substrate SAM, the TS and the product AZE in a strained conformation (Fig. 3c and Fig. S14). Interestingly, even in the absence of an enzymatic environment, this strain effect reduces the reaction barrier by ~6 kcal mol$^{-1}$ (Fig. 3a).

Moreover, our DFT calculations highlight the importance of charge transfer from sulfonium to C$^\gamma$H$_2$ and NH$_2$ during SAM conversion, which is modulated by specific residues (Figs. 3a, S16). A hydrophobic environment, as provided by Met28 and Leu136, facilitates charge transfer closer to the SAM carboxylate counter-ion (Fig. S14). The absence of water molecules in the AzeJ active site, as observed in the crystal structure, as well as the specific non-polar residues could thus contribute to the desolvation of the sulfonium and increase its reactivity to facilitate the cleavage of the C$^\gamma$-S bond. Indeed, a DFT cluster model of VioH, in which water molecules are bound near the sulfonium (Fig. S12), reveals a high reaction barrier of ca. 23 kcal mol$^{-1}$ in agreement with its low activity towards SAM (Fig. S17). Similarly, introducing a negative charge in close proximity to sulfonium as shown in the in silico mutant L136D destabilises the TS and the product by up to 5 kcal mol$^{-1}$ (Figs. 3a, S16). In the DFT models, Phe134 and Tyr175 are hydrogen-bonded to the NH$_2$ group of SAM or AZE, with Phe134 engaging in cation-π interaction additionally (Fig. S15). We thus hypothesise that these residues stabilise developing positive charge on the C$^\gamma$H$_2$ and NH$_2$ groups (Fig. S16).

Indeed, disrupting these interactions in the in silico mutants F134L and Y175F destabilises the TS and the product by up to 5 kcal mol$^{-1}$ (Figs. 3a, S16).

## Mutational analysis of AzeJ residues in catalytic function

To validate residues central to ligand binding and catalysis, a series of AzeJ variants was generated (Fig. S18 and Table S2). Our initial focus was on the hydrophobic environment of the sulfonium group. Mutations that increase polarity, such as L136D, L136N and M28N, are found to strongly affect activity, consistent with our desolvation hypothesis (Fig. 3b). Next, the importance of cation-π interactions between Tyr18 and the sulfonium was investigated. Indeed, the Y18A mutation resulted in a drastic reduction in catalysis, whereas Y18F retained about 60% activity compared to the wild-type (WT) protein (Fig. 3b). These results support the role of Tyr18 acting as an essential gating residue. Next, we characterised the function of Tyr175 and Phe134, which coordinate with the amino group of SAM and AZE. As expected, F134L and Y175A mutations resulted in loss of function. By contrast, F134H and Y175F retained 11–23% of the WT activity, whereas the conservative substitution F134Y performed as the WT (Fig. 3b). Kinetic parameters of F134Y and Y175F were further determined (Fig. 3d and Fig. S19). Both variants and AzeJ WT have similar apparent $K_M$ values, but the overall catalytic efficiency ($k_{cat}/K_M$) of F134Y and Y175F was reduced by 1.4- and 15-fold, respectively. To investigate the substrate binding capacity, thermal shift assays were performed for the WT, F134H and Y175A AzeJ variants, where the activity of the two latter was partially or completely abolished. In the presence of SAM or SAH, the melting temperature of AzeJ increased up to 8 °C. By contrast, these ligands had only a marginal stabilising effect on the two mutants (Fig. 3e). Thus, these data confirm the importance of the π and H-bonding interactions conferred by Phe134 and Tyr175 in the substrate- and/or product-bound state, and highlight a predominant role of Tyr175 during catalysis. Finally, the proposed role of Arg172 in proton transfer was investigated. As expected, substitution of this residue to Ala or Lys abolished the function, whereas the activity of R172Q was severely impacted (Fig. S20), suggesting that Arg172 could have a key function in mediating the proton transfer to the bulk solvent.

## Phylogenetic and genomic analyses of AZE synthases

To assess the taxonomic distribution of AZE synthases, a BLASTp search of AzeJ and VioH in the non-redundant protein sequences database of the NCBI was performed. This approach retrieved 375 homologues, clustered in 88 groups of proteins with >95% sequence identity (Supplementary Data 2). The 88 groups of homologues are sparsely distributed in bacteria and rarely in archaea (two genomes). Interestingly, they are mainly found in the phyla *Pseudomonadota* and *Actinomycetota*, with the next frequent occurrence in *Bacteroidota* and *Patescibacteria* (candidate phyla radiation). A remarkable portion of these microorganisms are from plant-associated microbiota, aquatic environments (e.g. wastewater, pond, lake), sludge and landfills. These analyses suggest that AZE production is not restricted to plants and may occur in unexpected environments.

A close inspection of the genomic context allowed to distinguish four major patterns of conservative genes around the *azeJ*-like gene, which correlate well with the topology of the phylogenetic tree of AzeJ (Fig. 4). Cluster I is clearly separated into two subgroups, each related to the biosynthesis of azetidomonamides or clipibicyclene natural products, respectively[8,18]. In contrast, the vioprolide BGC[9] is located in cluster III and contains genes encoding for a B12-dependent radical SAM (rSAM) enzyme, an aminoacyl-tRNA (AA-tRNA) synthetase, and occasionally NRPSs. This is suggestive of methyl-AZE as the main product, although it cannot be excluded that rSAM enzymes catalyse other modifications than methylation. For cluster II and IV, *azeJ*-like genes are grouped with those encoding enzymes for amino acid modification and peptide synthesis such as AA-tRNA synthetase or stand-alone NRPS adenylation domains. This indicates that the produced AZE would be further transformed and ligated into a peptide metabolite. Interestingly, clusters II-IV are frequently associated with elements that code for N-acetyltransferases, drug/metabolite transporters and/or α, β-hydrolases, all of which can be potentially involved in the detoxification of AZE[30–32]. Moreover, many clusters can be delineated by the presence of integrases, recombinases and transposases, suggesting that they are transferred by mobile genetic elements. Consistently, the rare occurrence of AzeJ across prokaryotes and the broad taxonomic diversity of *azeJ* encoding genomes is typical of an accessory gene that is spread through horizontal gene transfers rather than conserved through vertical inheritance[33,34]. Interestingly, AzeJ homologues from candidate phyla radiation bacteria form a distinct clade within the cluster III. As these microorganisms have a highly reduced genome due to the obligate symbiotic lifestyle[35,36], AZE-derived products may confer an advantage to their adaptation to the host.

## Incorporation of AZE into pyrrolizixenamides

Given that AZE can be recognised by the Pro-tRNA synthetase and subsequently incorporated into proteins[13,14,37], the question arises whether the NRPS Pro-specific adenylation (A) domain can activate AZE for peptide assembly. Therefore, we chose the pyrrolizixenamide pathway originated from *Xenorhabdus stockiae* as a model system for combinatorial biosynthesis, because it is closely related to that of azetidomonamides but lacks an AzeJ homologue[38]. Pyrrolizixenamides (2) are pyrrolizidine alkaloids, whose biosynthesis involves the assembly of acylated serine and proline by a bimodular NRPS (PxaA) followed by the action of a Baeyer-Villiger monooxygenase (PxaB) (Fig. 5a). Expression of *pxaA* alone in *E. coli* produced the intermediate indolizidine-2-one (1) (Fig. 5b). Feeding the culture with 1 mM AZE resulted in the production of a compound with the [M + H]$^+$ ion 251.13, identified as azetidopyridin-2-one (3), which was not detected in the absence of AZE (Fig. 5c). Although 3 is produced only in low yields (Fig. S23), this finding demonstrates the ability of the PxaA_A$_2$ domain to recognise AZE. Intriguingly, co-expression of *pxaA* with *azeJ* led to the production of 3, which is in contrast to the experiment with *pxaA* and *vioH* (Fig. 5c) and consistent with the in vitro observation that VioH alone had minimal catalytic activity (Fig. S8)[22]. Furthermore, when *pxaA*, *pxaB* and *azeJ* were expressed together, an azetidopyrroline analogue (4) of pyrrolizixenamide (2) was identified by tandem HR-MS analysis in the cell extract (Fig. 5d, e), albeit in trace amount. Thus, compounds 3 and 4 are azabicyclene analogues produced by heterologous expression[8,17,18], which opens up possibilities for the incorporation of proline mimics into natural products.

## Discussion

The electrophilicity of the sulfonium group in SAM activates all three adjacent alkyl groups, with their reactivity finely tuned by the enzyme environment. Here, we provide structural and mechanistic insights into AzeJ and VioH, the SAM-dependent AZE synthases, which exemplify this principle. The lyases catalyse the intramolecular cyclisation of the aminocarboxypropyl moiety of SAM to form AZE with a highly strained ring. They bind SAM in a kinked, near transition state conformation. This configuration facilitates an unusual 4-exo-tet cyclisation by positioning the nucleophilic α-amine and the C$^\gamma$-carbon at a distance of 2.9 Å.

SAM binding induces a conformational transition from an open to a closed state at the N-terminus of AzeJ, effectively removing solvent from the reaction chamber to suppress side reactions and to enhance the reactivity of the sulfonium group (Figs. S10, S11). Tyr18 likely serves as the gating residue, as supported by mutagenesis and activity assays. Intriguingly, our combined data suggest that SAM is bound with a neutral α-amine. This presumably results from a proton relay via a hydrogen bonding network involving Arg172 and Tyr176, in proximity to the carboxylate of SAM. The catalytic process most likely follows an

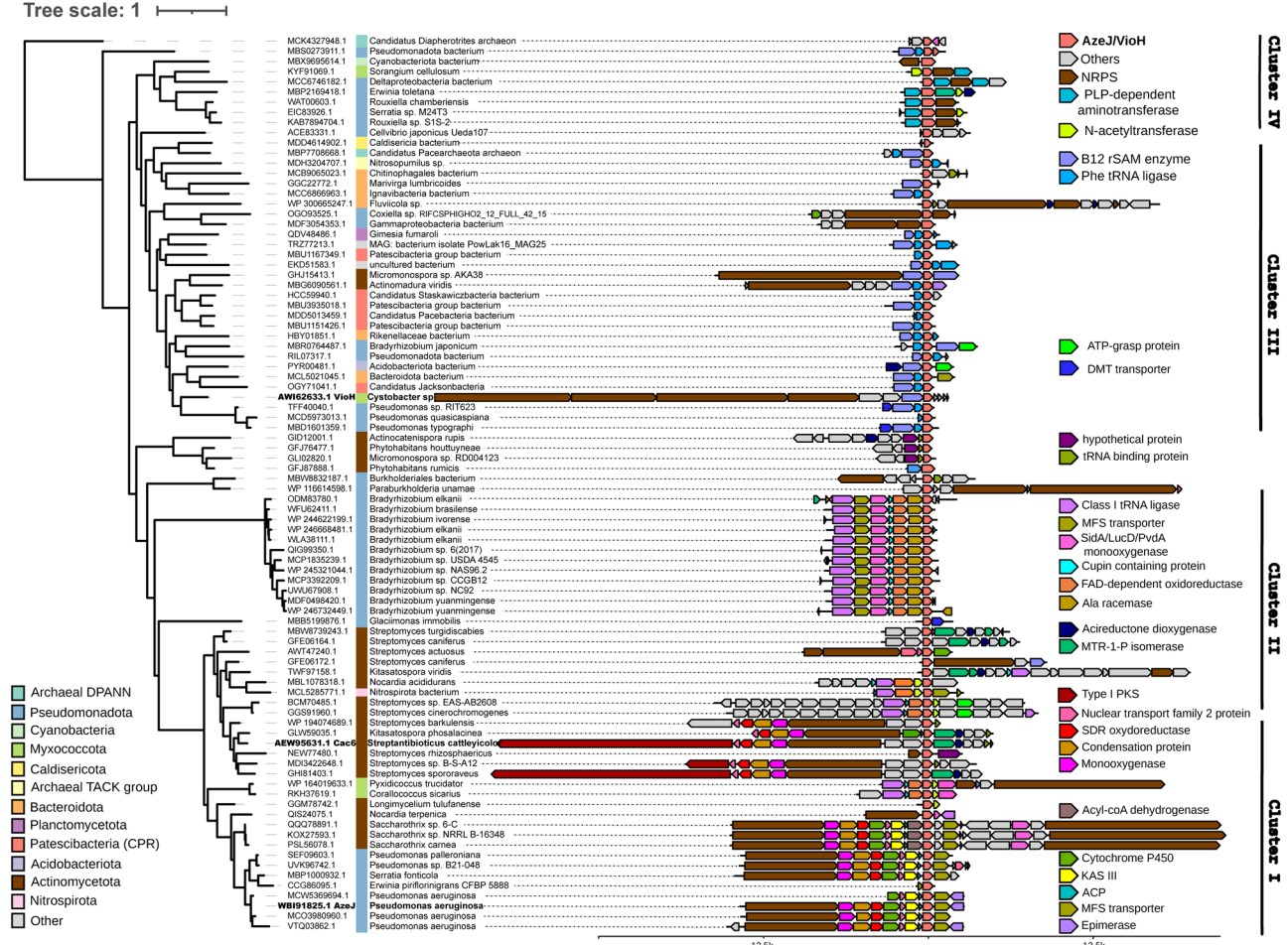

**Fig. 4 | Maximum likelihood phylogenetic tree and genomic context analysis of AZE synthases.** Accession numbers in bold highlight AzeJ, VioH and Cac6, involved in the biosynthesis of azetidomonamides[8], vioprolides[9] and clipibicyclene[18], respectively. The colour code illustrates relevant neighbouring genes encoding AzeJ-related products (coral pink), NRPSs (brown), B12-dependent rSAM enzymes (purple), class I tRNA-ligase (magenta), ATP-grasp enzymes (bright green), N-acetyltransferase (apple green) and drug/metabolite (DMT) transporter (cobalt). Source data are provided as a Source Data file.

$S_N2$ mechanism as commonly observed for enzymes belonging to the MT1 family[39,40], whereby the positive charge on the sulfonium transfers to the nitrogen atom of the formed azetidine ring and is stabilised by cation-π as well as hydrogen-bonding interactions (Figs. 2 and S16). An $S_N1$ mechanism is considered less favourable due to the instability of a primary $C^\gamma$ carbocation derived in such scenario. Alternative mechanisms include single electron transfer or covalent catalysis, for which we currently have no evidence. Of note, the reaction course catalysed by AzeJ involves several critical cation-π interactions, exemplified by the roles of Tyr18 and Phe134, highlighting the importance of such interactions in enzyme catalysis[41,42].

AzeJ and VioH, which catalyse the intramolecular cyclisation to form the azetidine ring are structurally distinct from other SAM lyases, such as (acyl)-HSL synthases and ACC synthases (Fig. 1a)[5,43,44]. The catalytic centres of AzeJ and the relevant phage HSL synthase Svi3-3[5] were aligned via the adenine residue of the surrogate SAH. Strikingly, the carboxylate group of HCY in Svi3-3 is positioned for nucleophilic attack on the $C^\gamma$ atom (distance 3.4 Å), whereas the amine group of HCY in AzeJ is perfectly aligned to form AZE (distance 2.9 Å) (Fig. S24). Taken together, this structural comparison illustrates how the precise arrangement of the nucleophile, electrophile and leaving group of SAM determines the reaction outcome, leading to the formation of either the γ-lactone or azetidine ring. In contrast, ACC synthases catalysing the 3-exo-tet cyclisation of SAM utilise a different mechanism. As demonstrated for plant enzymes[44–46], they require pyridoxal 5-phosphate as a cofactor to form a resonance-stabilised aldimine with the SAM-$NH_2$ group to deprotonate the $C^\alpha$-hydrogen. Furthermore, these lyases employ an active site Tyr to cleave the $C^\gamma$-S bond that assists closure of the three-membered cyclopropane ring. However, the different substrate orientation compared to AzeJ or HSL synthase is hampered by the lack of ACC synthase:SAH complex structures and awaits further studies for comparison.

AzeJ and VioH belong to the vast MT1 family, which encompasses members catalysing diverse SAM-dependent non-methylation reactions[1,2]. Here, the structural and biochemical data combined with multiple sequence alignment allow to propose sequence signatures that can distinguish AZE synthases from other MT1 enzymes. The highly conserved motifs $^{17}$FY$^{18}$, $^{133}$PFNxxGN$^{139}$, $^{172}$RxxYYxxC$^{179}$ (AzeJ numbering) are present in all AzeJ homologues (Fig. S21), where key roles of Tyr18, Phe134 and Tyr175 of AzeJ have been elucidated. Thus, the availability of AZE synthase structures now paves the way for further structure-based evolutionary studies[47].

Our large-scale bioinformatic studies identified dozens of AzeJ homologues in diverse bacteria phyla, which unveils a more prevalent production of AZE-related metabolites in nature. Since AZE is recognised by the machinery of protein synthesis, their presence in symbiotic, ultrasmall candidate phyla radiation bacteria are of particular interest to their ecological roles. On the other hand, the bioinformatic analysis reveals functional diversity of AZE synthases, providing promising avenues for natural product engineering.

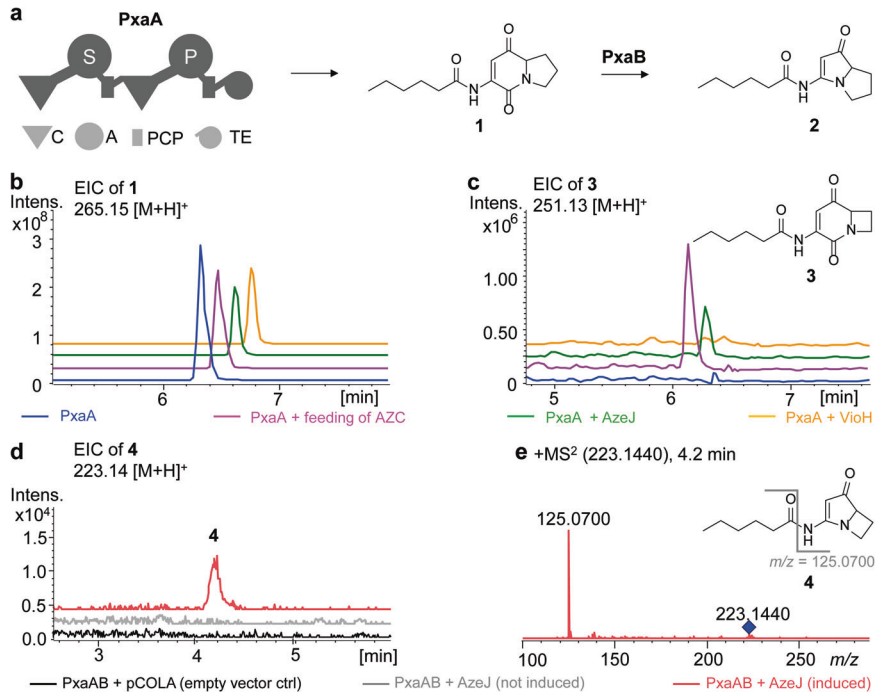

**Fig. 5 | Production of AZE-incorporated pyrrolizixenamides. a** Native biosynthetic scheme of pyrrolizixenamides involving the dimodular nonribosomal peptide synthetase PxaA together with the Baeyer-Villiger monooxygenase PxaB[38]. Domains: C (condensation), A (adenylation), PCP (peptidyl carrier protein), TE (thioesterase). Domain specificity is indicated in white. **b, c** Extracted ion chromatograms (EICs) of indicated compounds under various conditions: expression of *pxaA* in *E. coli* (blue), feeding AZE to a *pxaA*-expressing *E. coli* strain (pink), co-expression of *pxaA* and *azeJ* (green), as well as co-expression of *pxaA* and *vioH* (yellow) in *E. coli*. LC-HRMS analysis confirmed the identity of **1** ($m/z$ calc'd for $C_{14}H_{20}N_2O_3$ [M + H]$^+$ = 265.1546; obs'd. $m/z$ = 265.1548; Δppm = 0.3) and **3** ($m/z$ calc'd for $C_{13}H_{18}N_2O_3$ [M + H]$^+$ = 251.1390; obs'd. $m/z$ = 251.1390; Δppm = 0) (Fig. S23). Production of the azetidomonamide B (azabicyclene) analogue **4** upon co-expressing *pxaAB* and *azeJ* in *E. coli*: empty vector control (black), uninduced (grey), induced (red). The identity of **4** was confirmed by EIC (**d**) and by tandem mass spectrometry (**e**). Inset: proposed structure of **4** ($m/z$ calc'd for $C_{12}H_{18}N_2O_2$ [M + H]$^+$ = 223.1441; obs'd. $m/z$ = 223.1440; Δppm = −0.6). The key MS$^2$ fragment is indicated ($m/z$ calc'd for $C_6H_8N_2O$ [M + H]$^+$ = 125.0709; obs'd. $m/z$ = 125.0700; Δppm = −7.2).

AZE is a validated analogue of proline and its misincorporation into proteins by Pro-tRNA synthetases has been reported[12–14,37]. Similarly, the simultaneous identification of both AZE- and Pro-containing series of the same compound, as evidenced by azetidomonamides[8], vioprolides[9] and bonnevillamides[19,20], is indicative of the promiscuity of the corresponding NRPS A domains. Introducing an azetidine motif into the scaffold renders the molecule more rigid and might improve the pharmacological properties[11]. This study reveals that the pyrrolizixenamide NRPS Pro-specific A domain readily recognises AZE for peptide assembly. Simply providing AzeJ to the pathway enabled the synthesis of azabicyclene variants with a different acyl side chain in *E. coli*. Although NRPS engineering is likely required to increase the yield (e.g. by A domain engineering or exchange), this represents a prime example to generate azetidine-containing metabolites derived from NRPS pathways. Moreover, a recent study demonstrated that AZE is used by Ala-tRNA synthetases in vitro, based on its structural similarity to alanine[37]. We thus conclude that the potential of AZE synthases for combinatorial biosynthesis can also be applied to Ala-involved nonribosomal peptides.

## Methods

### Protein expression and purification

The genes encoding AzeJ (NP_252025.1) and VioH (AWI62633.1) were codon-optimised using SnapGene software and inserted into the pETDuet vector as N-terminal His$_6$-SUMO fusions. Site-directed mutagenesis (Tables S1 and S2) was performed using the Q5® Site-Directed Mutagenesis Kit (New England Biolabs), following the manufacturer's instructions and confirmed by Sanger sequencing (Eurofins Scientific).

Electrocompetent *E. coli* BL21 Gold (DE3) cells were transformed and cultured in 2 L glass flasks containing terrific broth (supplemented with 180 µg/mL ampicillin) at 37 °C. When the cultures reached an optical density at 600 nm (OD$_{600}$) of ~0.6, they were cooled at 4 °C for 30 min. Protein expression was induced by the adding of 1 mM isopropyl-β-D-thiogalactopyranoside (IPTG), and cells were incubated overnight at 20 °C. The resulting cell pellets were harvested by centrifugation, washed with 0.9% (w/v) NaCl, and stored at −20 °C.

For purification, 10 g of cells were resuspended in 50 mL of buffer A (100 mM Tris-HCl, pH 7.5; 500 mM NaCl; 20 mM imidazole) and lysed by sonication (Branson Digital Sonifier 250). Insoluble material was removed by centrifugation (40,000 × g, 4 °C, 30 min). The supernatant was then loaded at 5 mL min$^{-1}$ onto a 5 mL HisTrap HP column (ÄKTA Pure system, GE Healthcare) pre-equilibrated with buffer A. The column was washed with buffer A containing 5% buffer B (identical to buffer A but with 500 mM imidazole) until the absorbance of 280 nm stabilised at the baseline. The protein of interest was eluted by applying a linear gradient from 5% to 100% buffer B over a total volume of 50 mL. Fractions containing the target protein were pooled, supplemented with 0.5 mg SUMO protease (Ulp1 from *Saccharomyces cerevisiae*), and dialysed overnight at 4 °C against 2 L of buffer C (20 mM Tris-HCl, pH 7.5; 100 mM NaCl; 2 mM β-mercaptoethanol (β-ME)). The dialysed sample was reloaded onto the HisTrap column; the flow-through, containing the cleaved protein, was concentrated to 2 mL using Amicon Ultra-15 centrifugal filters. After clarification by centrifugation (20,000 × g, 4 °C, 10 min), the supernatant was applied to a HiLoad Superdex 75 16/60 column equilibrated in buffer D (identical to buffer C, with 2 mM dithiothreitol instead of β-ME) at a flow rate of 1.5 mL min$^{-1}$. The pure protein fractions were pooled and

concentrated to a minimum of 15 mg mL$^{-1}$. Samples were either used immediately for crystallisation experiments or flash-frozen and stored at −80 °C.

## Crystallisation, data collection, structure determination

Crystals of AzeJ and VioH were obtained by sitting drop vapour diffusion in Intelli 96-well plates (Art Robbins Instruments). Commercially available screens (Qiagen) were used to search for promising crystallisation parameters. Crystallisation screens were set up using the pipetting robot Phoenix (Art Robbins Instruments). Vapour diffusion droplets consisting of 0.2 µL protein mixed with 0.2 µL reservoir solution were prepared against 50 µL reservoir solutions on Intelli 96-well sitting-drop plates. Crystals were identified by using a transmission microscopy, and appeared within 7 days at 20 °C. In the presence of 2 mM SAM, AzeJ crystallised at two different conditions: 0.1 M HEPES, 2.4 M ammonium sulphate, pH 7.0 (space group P2$_1$2$_1$2) or 0.8 M sodium potassium phosphate, pH 7.2 (space group P4$_2$22). When supplemented with 2 mM SAH, AzeJ formed crystals in 1.8 M sodium potassium phosphate, pH 6.9 (space group P2$_1$2$_1$2), while VioH crystals formed in 1.4 M sodium potassium phosphate, pH 6.3 (space group P2$_1$2$_1$2). Cryoprotection of the crystals was performed by the addition of 1 µL mother liquor containing 30% (v/v) glycerol prior to vitrification in liquid nitrogen. Diffraction data were collected at the beamline X06SA, Swiss Light Source (SLS), Villigen, Switzerland. Reflection intensities were processed with the programme package XDS[48]. The structures were solved by molecular replacement with PHASER[49] using coordinates of the AlphaFold2[50] predictions as appropriate search models. After iterative model building and refinement steps with COOT[51] and REFMAC5[52], water molecules were placed automatically with ARP/wARP solvent[53]. Translation/libration/screw refinements finally yielded excellent values for R$_{crys}$ and R$_{free}$ as well as root-mean-square deviation (rmsd) bond and angle values. The models were proven to fulfil the Ramachandran plot using PROCHECK[54] (Table S4) and evaluated by MolProbity[55].

## AzeJ active site model and DFT calculations

Quantum chemical DFT cluster models of AzeJ and VioH were constructed based on the respective X-ray structures in complex with SAH (PDB IDs 8RYD and 8RYG, respectively). A summary of all models is given in Table S5 and Fig. S12 and their coordinates are provided in Supplementary Data 1. The DFT models of the active site comprised SAM (or MTA + AZE) and 14 surrounding amino acid residues ($N = 252$ atoms for AzeJ), as well as three water molecules in the case of VioH ($N = 304$ atoms). In addition, DFT calculations were also performed on the isolated ligand SAM (or MTA + AZE; $N = 49$ atoms) extracted from AzeJ models. AzeJ mutants were created in silico from the WT DFT model using Maestro[56]. Side chains were cut at the C$^\alpha$- or C$^\beta$-carbon, whereas backbone amides involved in hydrogen bonding were included. Terminal atoms were saturated and fixed during geometry and reaction pathway optimisations to account for steric effects of the protein environment. Amino acids were protonated according to their reference state based on PropKa estimations, as implemented in Maestro[56]. After initially considering a proton transfer from SAM's protonated HCY-amine (-NH$_3^+$) to the HCY-carboxylate (-COO$^-$) in AzeJ, this possibility was rejected based on the high energy and geometry of the resulting AZE-COOH product (Fig. S13). Thus, a neutral HCY-amine (-NH$_2$) was assumed for all models. For the reactant state, SAM was modelled from SAH by substitution to a methyl group in the appropriate S-configuration[3,4]. The product state was modelled from the geometry-optimised reactant state by adjusting the positions of the atoms in AZE based on the AzeJ:AZE:MTA X-ray structure (PDB ID: 8RYF).

Geometry optimisations were performed at dispersion-corrected DFT level, using the B3LYP-D3 functional and the def2-SVP basis set[57–60]. The protein surrounding in active site models was treated as a polarisable medium with a dielectric constant of $\varepsilon = 4$, using the COSMO implicit solvation model[61]. Isolated ligands (SAM or MTA + AZE) were solvated in dielectric media with low permittivity ($\varepsilon = 4$) or high permittivity ($\varepsilon = 80$), representing hydrophobic medium or water, respectively. Transition state structures were estimated based on reaction pathway optimisations using a zero-temperature string approach and showed a single imaginary frequency >400 cm$^{-1}$, consistent with the saddle point[62,63]. Single point energies for the reactants, products and transition states were calculated at the B3LYP-D3/def2-TZVP/$\varepsilon = 4$ level[64]. Energies were benchmarked against calculations at the B3LYP-D3/def2-QZVP/$\varepsilon = 4$ and ωB97X-D/def2-TZVP/$\varepsilon = 4$ levels. For isolated ligands, all calculations were additionally performed with a dielectric constant of $\varepsilon = 80$. Vibrational (including zero-point effects) and entropic corrections were obtained at the B3LYP-D3/def2-SVP level by estimating the Hessian numerically, using the resolution-of-the-identity (RI) approximation[65,66], a temperature of 298 K and a scaling factor of 0.9614. All electronic and free energies are given in Table S5. Strain effects were estimated based on the free energy difference of the protein optimised reactant/TS/product structures and the same structures, but just the isolated ligand, in water ($\varepsilon = 80$) or hydrophobic medium ($\varepsilon = 4$), as shown in Fig. S14. Charge distributions were derived from Mulliken charges. All QM calculations were performed with TURBOMOLE v. 7.5[67] and structures were visualised using Pymol[68].

## Small-scale purification of AzeJ variants for activity test

Cell pellets from 20 mL culture were resuspended in 5 mL lysis buffer (100 mM Tris-HCl, pH 7.5, 500 mM NaCl, 10 mM imidazole) and lysed by sonication. AzeJ variants were purified using HisPur Ni-NTA spin columns (1 mL resin bed, ThermoFisher Scientific) following the manufacturer's protocol and eluted into 4 mL buffer containing 100 mM Tris-HCl (pH 7.5), 500 mM NaCl and 250 mM imidazole. Proteins were subjected to buffer exchange using PD-10 columns (Cytiva) into the storage solution (100 mM Tris-HCl, pH 8.0, 150 mM NaCl and 10% glycerol), concentrated by Amicon ultra-centrifugal concentrators (cutoff 10 kDa) and flash frozen at −80 °C. Protein concentrations were determined by the Bicinchoninic Acid (BCA) protein assay kit (Pierce).

## Activity assays and kinetic determination

The concentration of the stock solution of SAM p-toluenesulfonate salt (Sigma-Aldrich) in deionised H$_2$O was determined by UV spectroscopy at 260 nm using the molar extinction coefficient 15,400 M$^{-1}$ cm$^{-1}$ [69]. The content of (S, S)-diastereomer of SAM was determined to be 79% by 1D-$^1$H NMR using a 600 MHz NMR spectrometer (Bruker)[70]. Enzyme activity assays were performed as previously described[8]. Briefly, a total of 40 µL reaction contained 100 mM Tris-HCl (pH 8.0), 1 mM SAM and 3 µM AzeJ or variants. The reaction was incubated at 30 °C for 1.5 h before being quenched by the addition of an equal volume of MeOH. MTA production was analysed on an HPLC Ultimate 3000 system (Thermo Scientific) equipped with a Luna HILIC column (5 µm, 200 Å, 250 × 4.6 mm, Phenomenex) and detected by UV absorbance at 260 nm. A gradient with buffer A (50% 50 mM ammonium acetate (pH 5.8)/50% acetonitrile) and buffer B (10% 50 mM ammonium acetate (pH 5.8)/90% acetonitrile) was applied: 100% B for 2.5 min followed by linear decrease to 0% B over 10 min, at a flow rate of 1 mL/min.

For kinetic measurements, AzeJ and variants were purified from 1 L culture and cleaved from the SUMO tag. Adenine stock solution was prepared in 0.5 M HCl and its concentration was determined by UV spectroscopy at 260 nm using the molar extinction coefficient 13,300 M$^{-1}$ cm$^{-1}$ [69]. The standard curve of adenine was generated by plotting HPLC peak areas. Assays were conducted in a total volume of 40 µL containing 100 mM Tris-HCl (pH 8.0), 0.5 µM MTA nucleosidase (G-Biosciences), 1–5 µM AzeJ or variants and varying concentration of SAM. A background control without protein was set up for each SAM concentration. Reactions were carried out at 30 °C for 10–20 min,

before being quenched by the addition of an equal volume of MeOH. Nine microliters of the mixture were subject to HPLC analysis. Data were measured in triplicates and analysed by Prism software (GraphPad). Curve fitting used the Michaelis-Menten model of the non-linear regression fitting function.

## Thermal shift assays

In a 25 μL reaction containing 100 mM Tris-HCl (pH 8.0), 10 μM AzeJ or its variants, 0–1 mM SAM or SAH, Sypro Orange protein gel stain (Merck) was added at a final dilution of 450-fold. Assays were performed in triplicates using a CFX96™ Real-Time System C1000™ Thermal Cycler (Bio-Rad) with a temperature gradient ranging from 20 to 80 °C and increments of 0.5 °C per 30 s.

## Phylogenetic and genomic context analysis

A BLASTp search of AzeJ and VioH was performed against the non-redundant protein sequence database, yielding 2645 hits[71]. Pairwise alignments of BLASTp were filtered according to the following cut-offs: Identity to VioH/AzeJ ≥ 27%, VioH/AzeJ and target length coverage ≥ 60%, E-value ≤ 1E-5, chosen so as to maximise the amount of reliable hits and minimise that of spurious ones (Fig. S22).

To remove redundancy from the dataset, a clustering of AzeJ, VioH and their 375 resulting high-confidence homologues was carried out using CD-HIT at an identity cut-off of 95%, yielding 88 clusters[72]. The representative protein of each cluster (Supplementary Data 2) was selected based on the criterion of being encoded by the longest contig. The 88 cluster representatives were then aligned using MAFFT with the parameters−maxiterate 1000−localpair for high accuracy[73]. The multiple sequence alignment was trimmed using trimAl with the -automated1 option optimised for maximum likelihood phylogeny inference[74]. The trimmed alignment of 179 sites was given as input to IQ-TREE to infer a maximum likelihood phylogenetic under the LG + G model of amino-acid substitution with 1000 ultrafast bootstraps[75]. To annotate the tree, genomic and taxonomic metadata of the 88 representative proteins were obtained from the NCBI protein database using efetch[76]. The genomic neighbourhood conservation analysis was performed as follows. The genomic region spanning from −20 kb to start codon to +20 kb from stop codon of each *azeJ/vioH* homologue (or less if the contig were smaller) was retrieved from the NCBI nucleotide database using efetch. Proteins encoded by the 88 regions were detected with Prodigal with the -m procedure optimised for gene calling on small contigs[77], and clustered into families of homologues by applying the MCL algorithm to the BLASTp all vs all alignment, with an inflating parameter set to 1.3[78]. The boundaries of the genomic vicinity of each *vioH/azeJ* homologue were manually narrowed to delineate regions of genes with low intergenic distances and enriched in protein families well conserved across the 88 contexts. The resulting refined genomic vicinities were plotted and mapped onto the leaves of the tree using the gggenomes R package (https://github.com/thackl/gggenomes) and further centred on the *azeJ/vioH* homologues. For each protein family, a relevance score was assigned as the number of occurrences of the protein family across the 88 contexts divided by the median distance (in number of CDS) of the family to *azeJ/vioH*. Families with a relevance score ≥ 2 were finally coloured and functionally annotated on the resulting figure. Specific features of surrounding genes were inspected and annotated manually.

## Construction of expression plasmids for pyrrolizixenamide pathway

For extraction of genomic DNA of *Xenorhabdus stockiae* DSM17904, the Monarch® Genomic DNA Purification Kit (New England Biolabs) was used according to the manufacturer's protocol for genomic DNA purification from Gram-negative bacteria. Plasmid DNA (Table S1) was isolated using the Monarch® Plasmid Miniprep Kit (New England Biolabs). DNA stretches of interest were amplified via PCR using primers with homology arms of ~25 bp obtained from Sigma-Aldrich (Table S3). The Q5® High-Fidelity DNA Polymerase (New England Biolabs) was used according to the manufacturers' protocol. DNA stretches were verified on a 1% (w/v) agarose gel and purified from agarose gels using the Monarch® DNA Gel Extraction Kit (New England Biolabs). Plasmid DNA was assembled using the NEBuilder® HiFi DNA Assembly Cloning Kit (New England Biolabs) following the manufacturer's protocol. The resulting DNA mix was then transformed into *E. coli* DH10B::*mtaA*[79] via electroporation using 1 mL cuvettes and the Gene-Pulser Xcell™ (Bio-Rad Laboratories GmbH) set at 1800 V, 25 μF, 200 Ω. Next, 500 μL Lysogeny broth (LB) medium was added and cells were incubated at 37 °C with shaking for 1 h. Cells were plated on LB agar with respective antibiotics and incubated overnight. For analytical purposes, DNA was digested using type II restriction enzymes supplied by New England Biolabs according to the manufacturer's protocol. The analytical digestions of plasmids were incubated at 37 °C for 1 h. All constructs were verified by Sanger sequencing (Microsynth).

## Heterologous production and LC-MS analysis of pyrrolizixenamides

For co-expression, *pxaA* from the pyrrolizixenamide gene cluster was cloned into a modified pACYC backbone under the control of pBAD promoter, alone or together with *pxaB*. The gene *azeJ* or *vioH* was inserted into a modified pCOLA backbone with a pBAD promoter (Table S1). A single colony of *E. coli* DH10B::*mtaA*[79] transformed with pACYC_pxaA or pACYC_pxaAB and pCOLA_azeJ or pCOLA_vioH was grown overnight in LB medium containing 50 μg/mL kanamycin and 34 μg/mL chloramphenicol. For production of desired peptides, 10 mL XPP medium[80] supplemented with 50 μg/mL kanamycin and 34 μg/mL chloramphenicol were inoculated with 100 μL of the pre-culture. *L*-Arabinose (2 μg/mL) was added at the beginning of the cultivation. Expression cultures were incubated at 22 °C with shaking at 200 rpm for 72 h. For crude extract preparation, 100 μL of the production culture was combined with 900 μL of methanol and incubated for 30 min. The mixture was centrifuged for 20 min at $14,000 \times g$. The cleared extract (100 μL) was subjected to liquid chromatography coupled to mass spectrometry (LC-MS) analysis. Liquid chromatography was performed on an UltiMate 3000 LC system (Thermo) with an installed C18 column (ACQUITY UPLC BEH C-18, 130 Å, 2.1 × 50 mm, Waters). Separation was conducted at a flow rate of 0.4 mL/min using a gradient of water containing 0.1% formic acid (v/v) (eluent A) and acetonitrile (eluent B) from 5 to 95% B over 18 min. Mass analyses were performed using an electrospray ionisation (ESI) mass spectrometer (AmaZon Speed, Bruker). For high-resolution MS analysis, LC-MS was performed on an Elute LC series 1300 (Bruker) with an installed C18 column (ACQUITY UPLC BEH C-18, 130 Å, 2.1 × 50 mm, Waters) and coupled to an ESI ion-trap mass spectrometer (timsTOF fleX MALDI 2, Bruker). The same LC separation gradient was used as described above. ESI-MS spectra were recorded in positive-ion-mode with the mass range from 100 to 1200 *m/z* and UV at 190–800 nm. The data were analysed applying the software Data Analysis 5.3 (Bruker).

## Reporting summary

Further information on research design is available in the Nature Portfolio Reporting Summary linked to this article.

# Data availability

Crystallographic data were deposited in the RCSB Protein Data Bank with the PDB identification numbers 8RYD, 8RYE, 8RYF and 8RYG. Source data are provided with this paper. Raw MS data are deposit in Zenodo (https://doi.org/10.5281/zenodo.14671958). Data supporting the findings of the study are available from the corresponding authors upon request. Source data are provided with this paper.

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

## Acknowledgements

We thank Benjamin Baetz and Caroline Giraud, University of Caen, for their assistance in thermal shift assays as well as the staff of beamline X06SA at the Paul Scherrer Institute, SLS, Villigen (Switzerland) for assistance during data collection. Financial support from the French National Research Agency (ANR), project no. ANR-20-CE92-0038-01 (to Y.L.) and the Deutsche Forschungsgemeinschaft (DFG, German Research Foundation), Project-ID 325871075–SFB 1309 (to M.G.) are gratefully acknowledged. V.R.I.K. acknowledges the Knut and Alice Wallenberg Foundation (KAW 2019.0251, KAW 2024.0220) and the Swedish Research Council for funding. The simulations were performed on resources provided by the National Academic Infrastructure for Supercomputing in Sweden (NAISS 2023/1-31, NAISS 2023/6-128, NAISS 2024/1-28) and the Leibniz Rechenzentrum (LRZ, project:pr83ro). We are grateful to Carsten Kegler (Bode lab) for providing the plasmids pCK_0431_pACYC_ara and pCK_0433_pCOLA_ara.

## Author contributions

H.B.B., Y.L. and M.G. initiated and supervised the project. T.J.K. and M.G. conducted crystallisation and analysed structural data. J.G. and V.R.I.K. performed quantum chemical calculations. C.B. conducted

bioinformatics analysis. J.E. performed heterologous expression experiments. C.L. and Y.L. performed and analysed biochemical studies. Y.L. and M.G. wrote the manuscript with contributions from all authors.

## Funding

## Competing interests

The authors declare no competing interests.
