## [Transparent Peer Review file · Nature Communications]

Molecular basis for azetidine-2-carboxylic acid biosynthesis

Corresponding Author: Professor Michael Groll

Version 0:

Reviewer comments:

Reviewer #1

(Remarks to the Author)

S-Adenosylmethionine (SAM) is a remarkably versatile intermediate in biochemistry and undergoes some important self-reactions beyond its roles as an alkyl donor and radical source (rSAM enzymes). The relatively less well-understood self-cyclization to azetidine-1-carboxylate is examined here from the perspective of x-ray crystallography of two enzymes that catalyze the cyclization reaction of SAM, DFT calculations of the reaction itself and extensive bioinformatics to establish how general the reaction is and how it plays parts in related natural product biosynthesis.

The paper is clearly written and comprehensive both historically and in its wider consideration of other SAM reactions. Focus on the actual mechanism of ring closure is central and joins mechanistic proposal with mutagenesis experiments that provide good indirect support for proposed roles of specific enzyme residues, e.g. Phe134, Tyr175 and Tyr18

For all its virtues, I have a fundamental problem with the mechanistic discussion starting on p. 4 and extending to the end. What their own data say to me quite clearly, but muddled in their presentation. First, they say on p. 9 "computational and biochemical data suggest that the amine group[ie. of the substrate, SAM] is already bound in a neutral state". Earlier on p. 4 it is said: "Strong cation- π interactions with Phe134 indicate a protonated state of the nitrogen atom." To be a nucleophile, the -NH₂ must be unprotonated. The organization of the substrate in the active site is kinked (pre-organized) for cyclization. Importantly, as noted by the authors, Tyr18 forms a strong pi-cation interaction with the sulfonium leaving group. Mutation Y18A destroys activity.

This is at base a simple SN₂ displacement of positively charged sulfonium and loss as the neutral MTA. The importance of Y18 is clearly understood by the authors. What is not, or is confusing to read, is the role of Phe134 and Tyr175 as pi-donors. They may "increase the nucleophilicity" of the nitrogen lone pair. I agree intuitively, but they do not give a good first-principles reason for the statement. Please visualize the following simple reaction coordinate: Substrate SAM binds and is stabilized by Tyr18, the attack of (neutral) nitrogen begins, partial positive charge is generated on the nitrogen as C-N bond formation begins to occur and that is stabilized by F134 and Y175 pi-cation (activation barrier is lowered), the C-S bond is breaking. The Y18 pi-sulfonium interaction is now disappearing and the leaving group is neutral in a hydrophobic environment, accelerating its departure. In the course of the reaction, developing + charge on nitrogen is stabilized by pi-cation interaction, existing + charge on sulfur is quenched as MTA is lost. Both contribute to lowering the transition state energy.

This is a good paper, but the authors should use their excellent data to sharpen the important central message of this study in a crisply written revised ms. The beautiful examples of pi-cation interaction deserve emphasis. SN₂ reactions are generally thought of as single TS processes

Small matters: the authors correctly state the absolute configuration in SAM at sulfur is (S). This fact is little appreciated and should be referenced (Cornforth, as I recall)

p.6, line 10 from bottom. The sentence should read: "...a series of AzeJ variants were [should be "series....was"]

p.8, middle of the page: "spread through horizontal gene transfers rather than conserved through vertical inheritance" A lot going on here that should be referenced.

Reviewer #2

(Remarks to the Author)

The goal of this study is to elucidate the structure and mechanism of azetidine-2-carboxylate (AZE) synthases. The AZE synthases are known enzymes, but the mechanism of strained azetidine ring formation remains a knowledge gap in the biosynthesis of this metabolite. The authors report liganded x-ray structures of two AZE synthases, AzeJ and VioH, bound to SAH, AZE, and MTA ligands in various combinations. The study provided valuable insight into the mechanism of azetidine ring formation including a structural view of a potential transition state in the AZE:MTA containing structure, which is supported by DFT TS calculations and biochemical characterization of active site mutants. The authors do some basic genome mining showing what is largely already known (these enzymes are found in a variety of microbial pathways, including NRPSs). The authors show that AZE can be incorporated into an NRPS pathway (pyrrolizixenamide) in competition with proline (just trace product formed to show it can be accepted as a substrate, this is not preparative scale production of a natural product analog).

The main criticism of the work is a lack of consideration of mechanistic alternatives to Sn2. How to rule out a single electron transfer pathway? Tyr (or Met/Cys) could mediate such a reaction pathway. The TS energy for the cyclization should be lower in a diradical coupling. The Y-to-F mutations do not effect Km but some do effect kcat. This is still consistent with a single electron transfer mechanism. Consider Sn2 vs Sn1. Also, consider covalent catalysis ... transfer of the SAM electrophile first to a Tyr or Cys side chain, then cyclization. It seems like some simple experiments such as a pH-rate profile might also provide useful insight. Need more consideration and discussion of alternate mechanisms in the discussion section. The authors seem fixated on Sn2, which is reasonable, but so are other mechanisms (Sn1, covalent catalysis, single electron transfer) that have not been ruled out.

Page 4: "proximity of the α -NH₂ group of HCY to Tyr175 and Phe134 ... enhance the nucleophilicity of the HCY nitrogen" ... this is speculation. Need to cite supporting literature or provide supporting experimental evidence to make this claim.

Page 5: "Based on the atomic insights, we propose that the sulfonium group acts as an electron acceptor" ... this is not a proposal based on structural observations, this is a simple fact of the bond making/breaking process.

Page 5: "Upon completion of the reaction, the positive charge migrates from the sulfonium to the AZE amine and is stabilised by cation- π interactions with Phe134 and Tyr175" ... more speculation. Need to cite supporting literature or provide supporting experimental evidence to make this claim.

Page 8: "the yield of 3 is lower than 1" How do you know this? There are no scale bars for the ion counts. The compounds will have different ionization potentials. Looks to me like you are only getting trace product formation here. Need to be transparent on interpreting the results (especially Figure 5e, where there is a high background and a small product peak in the pCOLA empty vector control). Please better explain and quantify.

Page 10: "Beta-Lactam ring" This is not a beta-lactam ring, it is an azetidine ring.

Figure 2: Panel 'c' only shows the nitrogen lone pair in some sort of orbital. This does not properly represent the reaction trajectory using frontier molecular orbital theory. Such a depiction requires showing the complete orbital (with back lobes) and showing the anti-bonding orbital of the electrophile.

Figure 3: The transition state structure is hard to visualize. The DFT calculations are not properly explained and there is much speculation and overreach in the mechanistic interpretation. How did the authors build this model ... starting from the AzeJ:SAH (VioH:SAH) conformation of SAH? Or, starting from the AzeJ:MTA:AZE structure? The orientation of the nitrogen nucleophile and carbon electrophile changes in these structures. It looks like the authors preloaded a conformation most consistent with the Sn2 pathway they were narrowly focused on. The curve fitting in panel 'd' is poor and is not sufficiently described. This brings into question the validity of the steady-state kinetic data as it really just looks like a straight line across. This needs improvement.

Reviewer #3

(Remarks to the Author)

The article "Molecular basis for azetidine-2-carboxylic acid biosynthesis" should be of interest to a broad audience including structural biologists, natural product chemists and engineers. The authors solve the structure of a representative bacterial azetidine-2-carboxylic acid (AZE) synthase and perform follow-up testing of their hypotheses with enzyme kinetics of mutants and computational chemistry calculations. Furthermore, they demonstrate the use of the AZE synthase for combinatorial biosynthesis by generating a novel pyrrolizixenamide derivative, which is a noteworthy achievement on its own. Together the authors tell a complete and coherent structure-function story about an interesting SAM using enzyme and its demonstrated use in combinatorial biosynthesis.

It's notable that the authors were unlucky enough to solve two different enzymes in the P21212 spacegroup, rather than the most likely P212121 spacegroup. The authors don't provide the redundancy of their data sets, but based on the ratio of observations to unique reflections most of them have decent redundancy. Since it is not reported, I'll assume the authors collected data on single crystals, rather than merging data from multiple data collections. The resulting R_{symm} (reported as R_{merge}) are about what are expected for classical cutoffs. The authors could easily go to higher resolutions based on the recommendations of Karplus and Diederichs. I encourage the authors to report and use modern cutoffs based on R_{pim}, CC*, CC1/2. See "Assessing and maximizing data quality in macromolecular crystallography" in Current Opinion in Structural Biology doi: 10.1016/j.sbi.2015.07.003

There aren't any signs that the authors are overinterpreting their data, and the authors make solid conclusions based on their structures. In some sense, there are no surprises that the enzyme uses substrate approximation and places the nucleophilic amine of SAM behind the electrophilic C-S bond, which is readily concluded by the product bound structures.

The authors seem to rightly struggle with the protonation state of the incoming SAM, as in solution the amine is expected to be protonated. It would be appreciated for the authors to include a table of the protonation states of the ligand and enzyme active site so that others can replicate their transition state calculations. Have the authors considered that the SAM carboxylate could accept the proton from the amine in the active site, such that it would be bound as the carboxylic acid? A pH rate profile would address that question if it was possible to deconvolute background decay of SAM from enzymatic activity at higher and lower pHs.

Is the difference between a C-N bond and a C-S⁺ bond really only -2.3 kcal/mol? The authors are missing an opportunity to state that there are two interesting high energy systems at play. It might be good for the authors to mention/discuss the relative energies of the strained four membered C-N bond compared to the methyl sulfonium ion in both the abstract and in the DFT calculations.

Suggested changes:

Page 2 – Abstract. “Besides, we uncover...” do the authors mean “In addition, we uncover...”

Page 5 – “determine the chemical energies” -> “calculate the chemical energies”

Page 10 – “rigid and would improve” -> “rigid and might improve” – such a modification is just as likely to stabilize a conformation disfavored for target engagement.

Reviewer #4

(Remarks to the Author)

Groll et al. investigated Molecular basis for azetidine-2-carboxylic acid biosynthesis using crystallization, mutagenesis analysis, quantum mechanical calculations, and phylogenetic and genomic analyses. The authors have employed a comprehensive approach to elucidate the mechanistic details of this important biochemical process. From an organic chemistry perspective, there are a few points that require further clarification. I recommend the publication of this manuscript in Nature Communication after addressing the following concerns in a major revision.

1. Figure 2c is incorrect. Under physiological conditions, amino groups are usually protonated, and it is unlikely that a lone pair of an amino group is exposed. In the case of intramolecular cyclization to form AZE, merely enhancing the leaving ability of S is insufficient; a mechanism to increase the nucleophilicity of the amine is necessary. Are there any such residues in AzeJ?

2. In Figure 2a and 2b, the main chain of F134 appears to form a hydrogen bond with the amino group of the substrate, but this seems insufficient. The pKa of a protonated carbonyl group is typically around -1.7, while the pKa of a protonated amino group is around 10.

3. In Figure 3c, Y175 appears to form a hydrogen bond with the amino group of the substrate. The pKa of a phenolic hydroxyl group is usually around 10, making it suitable for abstracting a proton from the amino group. However, according to Figure 3a, the activation energy does not change significantly even with Y175F. Please explain why this is the case.

4. What model was used for the DFT calculations in Figure 3a? I could not find any data related to the calculations in the Supporting Information. For a high-level journal like Nature Communications, the calculated coordinates, energies, magnitude of imaginary frequencies, and IRC plots should be provided.

5. It is not clear to the general reader whether the DFT calculations in Figure 3a are QM/MM or theozyme calculations. The term "theozyme calculation" should be used in the main text, and related references should be cited.

6. In Figure S11, is the structure optimization performed under any constraints? This point should be clearly stated.

Version 1:

Reviewer comments:

Reviewer #1

(Remarks to the Author)

The revised manuscript by Groll et al. is greatly improved with additional experiments and calculations carried out in response to extensive and useful reviewer comments. Questions and misgivings in the reviews center on the proposed mechanism and shortcomings in the data to support any defined mechanism. It is evident that considerable re-evaluation by the authors of the original questions and first mechanism have taken place. Despite appreciable additional work in response to the critiques, the new manuscript is interestingly more streamlined, clearer and direct reflecting better thinking and greater conviction, but at the same time it is more detailed and rigorous. The experiments and stronger data are in the end

persuasive. I agree with the proposed mechanism

Publication as it stands is merited

Reviewer #3

(Remarks to the Author)

The authors have addressed all the concerns I previously raised.

Reviewer #4

(Remarks to the Author)

The authors have adequately addressed all of the reviewers' questions. I believe the manuscript is suitable for publication in its current form.

Point-by-point response to referees' comments

We are grateful to all four referees for their time and constructive comments. We appreciate their feedback and the effort they have dedicated. Below, we address each suggestion for improvement point-by-point.

Referee #1

S-Adenosylmethionine (SAM) is a remarkably versatile intermediate in biochemistry and undergoes some important self-reactions beyond its roles as an alkyl donor and radical source (rSAM enzymes). The relatively less well-understood self-cyclization to azetidine-1-carboxylate is examined here from the perspective of x-ray crystallography of two enzymes that catalyze the cyclization reaction of SAM, DFT calculations of the reaction itself and extensive bioinformatics to establish how general the reaction is and how it plays parts in related natural product biosynthesis.

The paper is clearly written and comprehensive both historically and in its wider consideration of other SAM reactions. Focus on the actual mechanism of ring closure is central and joins mechanistic proposal with mutagenesis experiments that provide good indirect support for proposed roles of specific enzyme residues, e.g. Phe134, Tyr175 and Tyr18

1. For all its virtues, I have a fundamental problem with the mechanistic discussion starting on p. 4 and extending to the end. What their own data say to me quite clearly, but muddled in their presentation. First, they say on p. 9 "computational and biochemical data suggest that the amine group[ie. of the substrate, SAM] is already bound in a neutral state". Earlier on p. 4 it is said: "Strong cation- π interactions with Phe134 indicate a protonated state of the nitrogen atom." To be a nucleophile, the -NH₂ must be unprotonated.

We thank the reviewer for pointing out this apparent inconsistency. We have revised the sentence on p. 5 (line 10-11) to clarify that the "protonated state of the nitrogen" refers to the product AZE, not SAM. In addition, we have expanded the discussion on the deprotonated state of the SAM-NH₂ group once bound to AzeJ. This includes a possible deprotonation pathway via the SAM carboxylate and Arg172 (see Fig. S9). To further investigate this mechanism, we performed three additional mutation studies (R172A, R172K and R172Q, see Fig. S20), which supports that the network of hydrogen bonds involving Arg172 play a crucial role in the deprotonated state of SAM and the associated catalysis.

2. The organization of the substrate in the active site is kinked (pre-organized) for cyclization. Importantly, as noted by the authors, Tyr18 forms a strong pi-cation interaction with the sulfonium leaving group. Mutation Y18A destroys activity. This is at base a simple SN₂ displacement of positively charged sulfonium and loss as the neutral MTA. The importance of Y18 is clearly understood by the authors. What is not, or is confusing to read, is the role of Phe134 and Tyr175 as pi-donors. They may "increase the nucleophilicity" of the nitrogen lone pair. I agree intuitively, but they do not give a good first-principles reason for the statement. Please visualize the following simple reaction coordinate: Substrate SAM binds and is stabilized by Tyr18, the attack of (neutral) nitrogen begins, partial positive charge is generated on the nitrogen as C-N bond formation begins to occur and that is stabilized by F134 and Y175 pi-cation (activation barrier is lowered), the C-S bond is breaking. The Y18 pi-sulfonium interaction is now disappearing and the leaving group is neutral in a hydrophobic environment, accelerating its departure. In the course of the reaction, developing + charge on nitrogen is

stabilized by pi-cation interaction, existing + charge on sulfur is quenched as MTA is lost. Both contribute to lowering the transition state energy.

We thank the reviewer for this insightful suggestion. We have now streamlined the discussion of the mechanism following their recommendation and emphasized the impact of F134 and Y175. Our mutagenesis experiments and DFT calculations indicate that the π -character of F134 is important for catalysis, whereas Y175 stabilizes the transition state primarily through its hydroxyl group. This observation is in agreement with the finding that the Y175F mutation severely impairs activity compared to F134Y. Consequently, we revised the mechanistic chapter to better summarize how specific residues contribute to lowering the transition state energy according to the referee's suggestions.

3. This is a good paper, but the authors should use their excellent data to sharpen the important central message of this study in a crisply written revised ms. The beautiful examples of pi-cation interaction deserve emphasis. SN2 reactions are generally thought of as single TS processes.

The revised manuscript features a streamlined discussion of the proposed mechanism and emphasizes the critical role of cation- π interactions. We have also included the term "single transition state S_N2 reaction" in our DFT results.

4. Small matters: the authors correctly state the absolute configuration in SAM at sulfur is (S). This fact is little appreciated and should be referenced (Cornforth, as I recall)

In our revision the following references have been added: Cornforth et al. (DOI: 10.1038/2211212a0), Cornforth et al. (DOI: 10.1111/j.1432-1033.1970.tb00254.x).

p.6, line 10 from bottom. The sentence should read: "...a series of AzeJ variants were [should be "series....was"]

Done.

p.8, middle of the page: "spread through horizontal gene transfers rather than conserved through vertical inheritance" A lot going on here that should be referenced.

Two related references have been added as suggested.

Referee #2:

The goal of this study is to elucidate the structure and mechanism of azetidine-2-carboxylate (AZE) synthases. The AZE synthases are known enzymes, but the mechanism of strained azetidine ring formation remains a knowledge gap in the biosynthesis of this metabolite. The authors report liganded x-ray structures of two AZE synthases, AzeJ and VioH, bound to SAH, AZE, and MTA ligands in various combinations. The study provided valuable insight into the mechanism of azetidine ring formation including a structural view of a potential transition state in the AZE:MTA containing structure, which is supported by DFT TS calculations and biochemical characterization of active site mutants. The authors do some basic genome mining showing what is largely already known (these enzymes are found in a variety of microbial pathways, including NRPSs). The authors show that AZE can be incorporated into an NRPS pathway (pyrrolizixenamide) in competition with proline (just trace product formed to show it can be accepted as a substrate, this is not preparative scale production of a natural product analog).

1. The main criticism of the work is a lack of consideration of mechanistic alternatives to S_N2 . How to rule out a single electron transfer pathway? Tyr (or Met/Cys) could mediate such a reaction pathway. The TS energy for the cyclization should be lower in a diradical coupling. The Y-to-F mutations do not effect K_m but some do effect k_{cat} . This is still consistent with a single electron transfer mechanism. Consider S_N2 vs S_N1 . Also, consider covalent catalysis ... transfer of the SAM electrophile first to a Tyr or Cys side chain, then cyclization. It seems like some simple experiments such as a pH-rate profile might also provide useful insight. Need more consideration and discussion of alternate mechanisms in the discussion section. The authors seem fixated on S_N2 , which is reasonable, but so are other mechanisms (S_N1 , covalent catalysis, single electron transfer) that have not been ruled out.

We appreciate this suggestion. In line with Referee #1, we support an S_N2 mechanism but have included a brief discussion of alternative mechanisms (page 10, 2nd paragraph). Additionally, we have conducted activity assays of AzeJ at various pH values to support our mechanistic interpretation (see response to reviewer 3's question no.5). These results are now presented in Figure S25.

2. Page 4: "proximity of the α -NH₂ group of HCY to Tyr175 and Phe134 ... enhance the nucleophilicity of the HCY nitrogen" ... this is speculation. Need to cite supporting literature or provide supporting experimental evidence to make this claim.

We revised this statement to clarify that Phe134 forms amine- π interaction with SAM's α -amine. Despite less studied for such interaction (evidenced by reference 25-27), reference 28 would support our hypothesis that this interaction could increase the nucleophilicity of the nitrogen. The roles of Tyr175 and Phe134 are further discussed in the mechanistic chapter, including insights from mutagenesis data and DFT calculations.

2. Page 5: "Based on the atomic insights, we propose that the sulfonium group acts as an electron acceptor" ... this is not a proposal based on structural observations, this is a simple fact of the bond making/breaking process.

This sentence has been revised.

3. Page 5: "Upon completion of the reaction, the positive charge migrates from the sulfonium to the AZE amine and is stabilised by cation- π interactions with Phe134 and Tyr175" ... more

speculation. Need to cite supporting literature or provide supporting experimental evidence to make this claim.

This sentence has been revised accordingly. The cation- π interaction is well-known and identified in SAM-dependent methyltransferases, which is the basis of our hypothesis. Related references (e.g. reference 41) have been added. While the charge migration is a consequence of AZE formation (illustrated in Fig. S16), the effects of Tyr175 and Phe134 are now discussed in the context of evidence in the mechanistic chapter.

Page 8: “the yield of 3 is lower than 1” How do you know this? There are no scale bars for the ion counts. The compounds will have different ionization potentials. Looks to me like you are only getting trace product formation here. Need to be transparent on interpreting the results (especially Figure 5e, where there is a high background and a small product peak in the pCOLA empty vector control). Please better explain and quantify.

Our revised manuscript includes:

- Scale bars for ion counts were added to all chromatograms. We agree with the reviewer that different compounds can have different ionization potential in the mass spectrometry. However, given the near identical structure between compound **1** and **3**, we reasoned that they should have very similar ionization behavior, hence the mass intensity can give us a qualitative comparison of the yield.
- The small peak in the pCOLA empty vector control in the original figure is derived from non-specific background. To improve the chromatogram quality, we repeated MS analysis with more concentrated extracts. New chromatograms (see Fig. 5d) were added showing a clear peak for compound **4**, albeit at low intensity, and no product peak in the negative controls (pCOLA empty vector or AzeJ un-induced).
- A new supplementary Fig. S23 was added to show the relative quantification of AZE- vs. Pro- incorporated pyrrolizinenamides.
- We now state that compound **4** is produced in trace amounts.

4. Page 10: “Beta-Lactam ring” This is not a beta-lactam ring, it is an azetidene ring.

Beta-Lactam ring has been corrected to azetidene ring.

5. Figure 2: Panel ‘c’ only shows the nitrogen lone pair in some sort of orbital. This does not properly represent the reaction trajectory using frontier molecular orbital theory. Such a depiction requires showing the complete orbital (with back lobes) and showing the anti-bonding orbital of the electrophile.

We have revised Fig. 2c as suggested, while keeping it simple for clarity and readability.

6. Figure 3: The transition state structure is hard to visualize. The DFT calculations are not properly explained and there is much speculation and overreach in the mechanistic interpretation. How did the authors build this model ... starting from the AzeJ:SAH (VioH:SAH) conformation of SAH? Or, starting from the AzeJ:MTA:AZE structure? The orientation of the nitrogen nucleophile and carbon electrophile changes in these structures. It looks like the authors preloaded a conformation most consistent with the S_N2 pathway they were narrowly focused on. The curve fitting in panel ‘d’ is poor and is not sufficiently described. This brings into question the validity of the steady-state kinetic data as it really just looks like a straight line across. This needs improvement.

We have updated Fig. 3c and now additionally provide an alternate view of the transition state in Fig. S16c. We have completely revised the Methods section for the DFT calculations and included a brief explanation of the model building in the main text. The geometries of reactants and products (Fig. S15) were optimized based on the crystal structures and represent local minima as demonstrated by the lack of imaginary frequencies. Moreover, the transition state obtained from reaction pathway optimizations (Fig. 3c) represents a saddle point with a single imaginary frequency corresponding to the reaction coordinate. According to the DFT cluster models, the S_N2 reaction has a kinetically feasible free energy barrier of ca. 16.6 kcal mol⁻¹, which has been clarified in the revised version. Furthermore, we have included that the kinetic curve fitting was performed by GraphPad using the Michaelis-Menten model of the nonlinear regression analysis. This detail and the R² value of each curve were added to the methods section and corresponding figure legends (Fig. S19). For Y175F, a reliable curve was obtained without consideration of the substrate inhibition effect, which allowed us to calculate apparent kinetic parameters. Both fitted curve and raw data of Y175F were added to Fig. S19.

Referee #3:

The article “Molecular basis for azetidine-2-carboxylic acid biosynthesis” should be of interest to a broad audience including structural biologists, natural product chemists and engineers. The authors solve the structure of a representative bacterial azetidine-2-carboxylic acid (AZE) synthase and perform follow-up testing of their hypotheses with enzyme kinetics of mutants and computational chemistry calculations. Furthermore, they demonstrate the use of the AZE synthase for combinatorial biosynthesis by generating a novel pyrrolizixenamide derivative, which is a noteworthy achievement on its own. Together the authors tell a complete and coherent structure-function story about an interesting SAM using enzyme and its demonstrated use in combinatorial biosynthesis.

1. It's notable that the authors were unlucky enough to solve two different enzymes in the P21212 spacegroup, rather than the most likely P212121 spacegroup. The authors don't provide the redundancy of their data sets, but based on the ratio of observations to unique reflections most of them have decent redundancy.

We have expanded Supplementary Table S4 and added the redundancy. As expected by the reviewer, we have a solid multiplicity of at least 3.6.

2. Since it is not reported, I'll assume the authors collected data on single crystals, rather than merging data from multiple data collections.

Please see Supplementary Table S4: [c] Data reduction was carried out with XDS and from a single crystal. Friedel pairs were treated as identical reflections. We highlighted this passage in our revised version for easy viewing.

3. The resulting R_{sym} (reported as R_{merge}) are about what are expected for classical cutoffs. The authors could easily go to higher resolutions based on the recommendations of Karplus and Diederichs. I encourage the authors to report and use modern cutoffs based on R_{pim}, CC*, CC1/2. See “Assessing and maximizing data quality in macromolecular crystallography” in Current Opinion in Structural Biology doi: 10.1016/j.sbi.2015.07.003.

There aren't any signs that the authors are overinterpreting their data, and the authors make solid conclusions based on their structures. In some sense, there are no surprises that the enzyme uses substrate approximation and places the nucleophilic amine of SAM behind the electrophilic C-S bond, which is readily concluded by the product bound structures.

We appreciate that our data could even be processed at higher resolutions. However, processing our data at higher resolution does not improve the quality of the electron density maps and therefore, we do not obtain any additional information about the mode of action. Thus, we would like to continue presenting our data sets in a more conservative way.

4. The authors seem to rightly struggle with the protonation state of the incoming SAM, as in solution the amine is expected to be protonated. It would be appreciated for the authors to include a table of the protonation states of the ligand and enzyme active site so that others can replicate their transition state calculations.

We have included a new Table S5, which lists the protonation states of the HCY-amine used in each model. Active site residues were protonated according to their dominant protonation state in solution. For better reproducibility, we now provide the geometry-optimized coordinates of reactants and products for all models in Table S6.

5. Have the authors considered that the SAM carboxylate could accept the proton from the amine in the active site, such that it would be bound as the carboxylic acid? A pH rate profile would address that question if it was possible to deconvolute background decay of SAM from enzymatic activity at higher and lower pHs.

We thank the reviewer for this excellent idea. We had originally tested a proton transfer to the SAM carboxylate explicitly in DFT calculations (Fig. S13). Although this transfer is kinetically feasible, the reaction is uphill by ca. 9 kcal mol⁻¹. The AzeJ:SAH-complex depicts the carboxylate group of the ligand H-bonded to N139N^{δ2}, S203O^γ and Y176OH that favors COO⁻. We thus concluded that R172 is of key importance in gating the proton transfer to bulk solvent. Notably, R172 is strongly coordinated to S203O^γ and Y176OH in the determined structure (see Fig. S9), but could undergo a conformational change similar as many other systems. We therefore replaced Arg172 with Ala or Lys, which resulted in a complete loss of function, whereas the R172Q mutation caused a significant reduction in activity (see Figure S20), supporting that Arg172 is functionally important.

As suggested, we attempted to conduct a pH-rate profile experiment of pH vs log k_{cat}/K_M . However, as the reviewer pointed out, SAM degraded rapidly at higher pH (> 9.5), preventing reliable kinetic measurements using our HPLC-based end-point assays. In addition, the ionization state of SAM varies with the pH value, complicating data interpretation and precluding unambiguous conclusions (REF: Tipton KF and Dixon HB. *Methods Enzymol.* 1979 (doi: 10.1016/0076-6879(79)63011-2)). Nonetheless, we observed that AzeJ exhibits its maximum initial rate around pH 9, which is close to the pK_a of SAM's amine group (see Fig. S25). Based on these findings, we have narrowed down the possible deprotonation pathways and hypothesise that the proton is transferred either to the bulk solvent in the open state via pK_a modulation of the active site or a hydrogen bonding network including Arg172.

Is the difference between a C-N bond and a C-S⁺ bond really only -2.3 kcal/mol? The authors are missing an opportunity to state that there are two interesting high energy systems at play. It might be good for the authors to mention/discuss the relative energies of the strained four membered C-N bond compared to the methyl sulfonium ion in both the abstract and in the DFT calculations.

We thank the reviewer for this suggestion. We now report the free energy difference, including vibrational and entropic corrections, which is even smaller (ca. -1 kcal mol⁻¹). This reflects not only the difference in bond energy, but also other factors such as electrostatic interactions (e.g., the positive charge moving closer to the carboxylate anion), strain from the azetidinium ring, and complex interactions with the active site. Notably, a similarly small free energy difference is observed in reaction models in water, indicating that this balance, though reduced, persists even in the absence of AzeJ. We have expanded on the small energy difference between the high-energy states, as well as the discussion of the DFT calculations, in our revision.

6. Suggested changes:

Page 2 – Abstract. “Besides, we uncover...” do the authors mean “In addition, we uncover...”

Done.

Page 5 – “determine the chemical energies” -> “calculate the chemical energies”

Done.

Page 10 – “rigid and would improve” -> “rigid and might improve” – such a modification is just as likely to stabilize a conformation disfavored for target engagement.

Done.

Referee #4:

Groll et al. investigated Molecular basis for azetidine-2-carboxylic acid biosynthesis using crystallization, mutagenesis analysis, quantum mechanical calculations, and phylogenetic and genomic analyses. The authors have employed a comprehensive approach to elucidate the mechanistic details of this important biochemical process. From an organic chemistry perspective, there are a few points that require further clarification. I recommend the publication of this manuscript in Nature Communication after addressing the following concerns in a major revision.

1. Figure 2c is incorrect. Under physiological conditions, amino groups are usually protonated, and it is unlikely that a lone pair of an amino group is exposed. In the case of intramolecular cyclization to form AZE, merely enhancing the leaving ability of S is insufficient; a mechanism to increase the nucleophilicity of the amine is necessary. Are there any such residues in AzeJ?

We greatly appreciate the reviewer's comment. Indeed, under physiological conditions, the α -amine of SAM is predominantly protonated (pKa for the methionine amine is ~ 9.3). However, it is well-established that the protein environment can shift the pKa by up to 3 units, potentially allowing the amine to remain neutral when SAM is bound to AzeJ. To this end, we have systematically investigated possible deprotonation pathways. Initially a deprotonation via Tyr175 and Cys179 was suspected, but the mutagenesis excluded this possibility. Likewise, a proton transfer to the SAM carboxylate is feasible but not sufficient for initiating the cyclisation according to DFT calculations (see Fig. S13). Therefore, we carefully reanalyzed the interaction network around SAM and could identify Arg172 as a key residue in catalysis. The proposed role of Arg172 in proton transfer was further investigated by mutagenesis and activity assay studies. Strikingly, the replacement of Arg172 with Ala or Lys resulted in a complete loss of function, while the R172Q mutation caused a significant reduction in activity (see Figure S20). We thus hypothesise that the R172 is central for gating the proton transfer to bulk solvent after binding to AzeJ (see also Fig. S9). Moreover, AzeJ exhibits highest initial rate at a pH near the pKa (8.5-9.5), which supports our hypothesis that the proton is transferred to bulk solvent after SAM binds to AzeJ. Taken together, our new findings have narrowed down a possible mechanism of amine deprotonation.

2. In Figure 2a and 2b, the main chain of F134 appears to form a hydrogen bond with the amino group of the substrate, but this seems insufficient. The pKa of a protonated carbonyl group is typically around -1.7, while the pKa of a protonated amino group is around 10.

Our activity assays and AzeJ:AZE:MTA complex structure show that SAM is cyclized to AZE by AzeJ at pH 7. Since this conversion requires a neutral amine, SAM must be in a deprotonated state when bound to AzeJ. In our revised version, we discuss potential deprotonation pathways (see previous response), and clarify that the NH_3^+ -group of SAM is not deprotonated via the carbonyl group of F134O.

3. In Figure 3c, Y175 appears to form a hydrogen bond with the amino group of the substrate. The pKa of a phenolic hydroxyl group is usually around 10, making it suitable for abstracting a proton from the amino group. However, according to Figure 3a, the activation energy does not change significantly even with Y175F. Please explain why this is the case.

We now provide a detailed discussion of the protonation state on the reactants and the protein residues involved. We further clarify that proton transfer to Y175, which would result in two adjacent cations upon conversion to AZE (positively charged amine), is thermodynamically

disfavored. Additionally, proton transfer from Y175 to C179, the nearest potential proton acceptor, is also unlikely, as demonstrated by the C179A mutant, which retains full catalytic activity as observed in wild-type AzeJ. Furthermore, the observed change in activation energy for the Y175F mutant of about 2.1 kcal mol⁻¹ corresponds to a substantially slower reaction rate (ca. 40-fold) as predicted by transition state theory, which is consistent with the residual activity of the Y175F mutant. We revised our manuscript accordingly.

4. What model was used for the DFT calculations in Figure 3a? I could not find any data related to the calculations in the Supporting Information. For a high-level journal like Nature Communications, the calculated coordinates, energies, magnitude of imaginary frequencies, and IRC plots should be provided.

We appreciate the referee's comment and have revised the methods section as well as the main text about DFT calculations to increase clarity and transparency. We have clarified the details of the DFT cluster models from the AzeJ:SAH structure both in the main text as well as the methods section. We have included vibrational and entropic corrections and now report free energies which allow a better comparison with experiments. Moreover, a new Table S5 gives an overview over all models and includes information about the protonation states, total charge as well as electronic and free energies. The accessibility and reproducibility is further enhanced by the new Table S6, providing geometry-optimized coordinates of the models. We have ensured the quality of our models by checking that reactant and product states represent energy minima by analysis of the molecular Hessian and that transition states represent saddle points (i.e. contained a single imaginary frequency of ~ 400-450 cm⁻¹ corresponding to the reaction coordinate). Intrinsic Reaction Coordinate (IRC) plots of the reaction pathways optimized at the B3LYP-D3/def2-SVP level showed clearly defined single transition states.

5. It is not clear to the general reader whether the DFT calculations in Figure 3a are QM/MM or theozyme calculations. The term "theozyme calculation" should be used in the main text, and related references should be cited.

We used the well-established concept of DFT cluster models of the active site and have revised our manuscript accordingly.

6. In Figure S11, is the structure optimization performed under any constraints? This point should be clearly stated.

We have emphasized the use of fixed atoms in the figure description (now Fig. S12), and now provide detailed information on their coordinates in Table S6.